

# Flexural properties of 3D printed continuous flax fiber reinforced polyethylene composites

Minggan Wang[1,2,3], Xiaohui Song[2,3], Zhengwei Yang[2,3], Chunlei Luo[2,3], Songda Chi[2,3] and Hao Jiang[2,3]

[1] College of Mechanical and Control Engineering, Guilin University of Technology, Guilin, Guangxi Zhuang Autonomous Region, China

[2] Key Laboratory of Special Engineering Equipment Design and Intelligent Driving Technology (Guilin University of Aerospace Technology), Education Department of Guangxi Zhuang Autonomous Region, Guilin, Guangxi Zhuang Autonomous Region, China

[3] Guangxi Key Laboratory of Special Engineering Equipment and Control, Guilin University of Aerospace Technology, Guilin, Guangxi Zhuang Autonomous Region, China

Corresponding author
Xiaohui Song,
songxiaohui2010@163.com

## ABSTRACT

Natural fibers are classified as green energy materials and have embodied the principles of sustainable development. This paper impregnated natural continuous flax fibers (CFF) with polyethylene (PE) on a self-developed melt impregnation device, obtained CFF/PE prepreg filaments. A fused deposition machine was used to 3D print CFF/PE composites. The effect of impregnation temperature on tensile properties of CFF/PE filaments was explored. The influence of various 3D printing parameters on bending performance and dimensional accuracy of CFF/PE composites was investigated. The results indicated that the tensile strength of the prepreg filament reached to the maximum value (12.41 MPa) at impregnation temperatures of 150 °C–155 °C. At a layer width of 1.5 mm, a layering thickness of 0.8 mm, a printing speed of 10 mm/min, and a printing temperature of 200 °C, the flexural strength of CFF/PE composites was improved to 24.74 MPa from 13.14 MPa. When the layer width was 2.0 mm and the layer thickness was 0.6 mm, the porosity was reduced to 3.88%. At a layer width of 1.5 mm, severe polymer accumulation occurred between adjacent printing lines, resulting in a significant impact on the dimensional accuracy of the composites. The maximum deviation was approximately 1.45 × 0.57 × 2.32 (mm). As the number of bendings increased, the bending rebound angle decreased, and the bending performance deteriorated, until it kept level.

## INTRODUCTION

Fiber-reinforced polymers have gained extensive application and attention in high-end manufacturing sectors such as aerospace and automotive industries due to their corrosion resistance, light weight, and exceptional mechanical performance (*Shan et al., 2020*; *Zhang et al., 2023*). Traditional composite manufacturing methods normally adopted molds and jigs resulting in high cost, longer production cycle, and limited design flexibility (*Matsuzaki*

*et al., 2016*; *Tian et al., 2021*). Therefore, it is very difficult to manufacture complex parts and to satisfy the requirements of individual customization. Fused deposition modeling (FDM) represents one of the 3D printing techniques. Being compared to other 3D printing methods, FDM has emerged as one of the most widely utilized 3D printing technology (*Daminabo et al., 2020*; *Vyavahare et al., 2020*) due to its easier operating, and lower price. Materials used on FDM include polylactic acid (PLA), nylon (PA), and polyether ether ketone (PEEK), etc. The incorporation of fibers significantly enhanced the mechanical properties of these polymer materials or achieved desired functionalities (*Cao et al., 2023*; *Tan et al., 2023*). Moreover, continuous fibers exhibited superior mechanical property enhancement over short fibers (*Yu, Nie & Luo, 2023*), thus obtained increasing focus in recent years.

The continuous fiber-reinforced composites studied by these scholars primarily utilized synthetic fibers such as carbon fiber, glass fiber, and aramid fiber. *Yang et al. (2017)* utilized continuous carbon fibers (CCF) to reinforce PLA, employed the in-situ impregnation process to fabricate CCF/PLA composites. They found that process parameters such as fiber content, printing temperature, and pressure significantly influenced the properties of composites. *Zhang et al. (2019)* produced CCF/PLA prepreg filaments and revealed that the mechanical properties of the composites were optimized at a printing temperature of 200 °C and printing speed of 300 mm/min. *Mathur, Kabir & Seyam (2020)* investigated the effect of print parameters on the tensile properties of continuous glass fiber (CGF)/PA composite specimens and showed that the tensile properties were mainly affected by the fiber orientation. *Cersoli et al. (2021)* found that the tensile properties of Kevlar fiber reinforcement PLA composites with a fiber volume fraction of 20.53% were more than double that of pure PLA. These artificial fibers are generally difficult to degrade and have high manufacturing costs (*Mochane et al., 2019*). Due to the increasingly severe environmental and energy issues, the research on sustainable green energy materials has gained significant focus in recent years. Composites reinforced with natural fibers also possessed high comprehensive performance and offered advantages such as complete biodegradability, renewability, easy availability, and low cost (*Ilyas et al., 2022*; *Zhan, 2021*). Common natural fibers used as reinforcement include jute fiber, flax fiber, ramie fiber, and sisal fiber. *Matsuzaki et al. (2016)*, *Long et al. (2023)*, *Long et al. (2024)*, *Le Duigou et al. (2020)*, *Cheng et al. (2021)* and *Zhu (2017)* respectively utilized jute fiber, flax fiber, ramie fiber, and sisal fiber to reinforce polymer. The results consistently indicated that natural continuous fibers as reinforcement materials significantly improved the mechanical properties of the polymers.

Polyethylene (PE) is widely used due to its corrosion resistance, low-temperature tolerance, high ductility, and stable chemical properties (*Wang, Huo & Yang, 2020*). *Chong et al. (2016)* demonstrated that the feasibility of using recycled high-density polyethylene (HDPE) as a 3D printing raw material. *Schirmeister et al. (2019)* utilized fused filament fabrication (FFF) to produce HDPE filaments, and improved the mechanical properties and surface quality of composites by adjusting processing parameters. Although the optimization of process parameters can enhance the mechanical properties of PE, fiber-reinforced PE composites exhibited superior mechanical performance. *Borkar et al. (2022)*

incorporated carbon fibers (CF) into HDPE, and found that higher CF content increased tensile and flexural strength of the 3D printed specimens, with maximum values of 39.9 MPa and 53.9 MPa, respectively. *Koffi et al. (2021)* used short yellow birch fibers to reinforce HDPE, the composite's Young's modulus increased by 35% at 30 wt% yellow birch fibers compared to pure HDPE. However, due to the good fluidity and high shrinkage rate of PE, it is difficult to form into filament under normal conditions, and this can lead to deformation of the printed parts, affecting the dimensional accuracy of the prints. Continuous fiber reinforcement of PE can effectively address these issues. Currently, there is a scarcity of literature on the 3D printing PE reinforced with CFF. Therefore, this study will continue to explore this area of research.

This study employed polyethylene (PE) as the matrix material and flax yarn as the continuous fiber reinforcement. The main contents were focused: using a self-developed single-screw extruder impregnation device to explore the effects of extrusion temperature and impregnation temperature on the properties of continuous fiber prepreg filaments; using the prepreg filaments as raw material to investigate the influence of different process parameters under FDM technology on flexural properties, porosity, and dimensional accuracy; conducting five flexural tests to examine the impact law of the number of tests on flexural angle and performance.

## MATERIALS AND METHODS

### Raw materials

Polyethylene (PE) powder (density of 0.93 g/cm$^3$, 100 mesh) was provided by Dongguan Huaxin Plastic Co., Ltd., China. The continuous flax fiber yarns (density of 1.5 g/cm$^3$, linear density of 68Tex) were obtained from Shanxi, China. The flax fibers and PE powder were dried in an oven at 60 °C before being used.

### Experimental methods

#### Prepreg filaments production

Continuous flax fiber (CFF)/PE prepreg filaments were prepared using a self-developed single-screw extruder. The experimental principle and impregnation equipment (*Song et al., 2024*) is shown in Fig. 1. Within the barrel, PE was heated to a molten state and then was transported to the impregnation mold under the action of the extrusion screw. The PE melt was kept at the melting state in the mold. The CFF (Fig. 2A) moved along the direction of traction, passed through the impregnation mold, and was impregnated with PE melts under high pressure within the mold. The impregnated filaments were subsequently drawn out through the mold. Therefore, prepreg CFF/PE filaments with 1.6 mm diameter were obtained (Fig. 2B). Table 1 shows the impregnation parameters. The extrusion speed, traction speed, and die diameter were fixed as 8 r/min, 300 mm/min and 1.5 mm, respectively. The extrusion temperature (T1, the barrel temperature) and the impregnation temperature (T2, the mold temperature) were variables to explore the effect of temperature on filament properties.

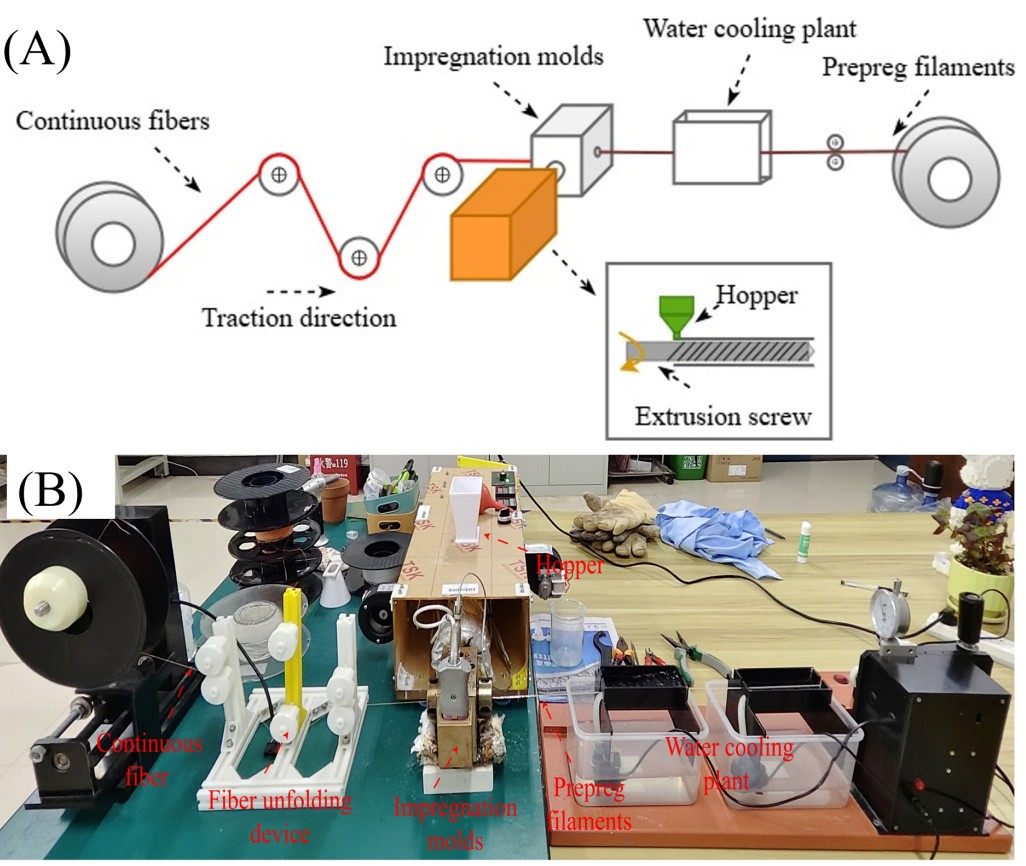

**Figure 1** Prepreg CFF/PE filaments: (A) Experimental principle; (B) impregnation equipment.

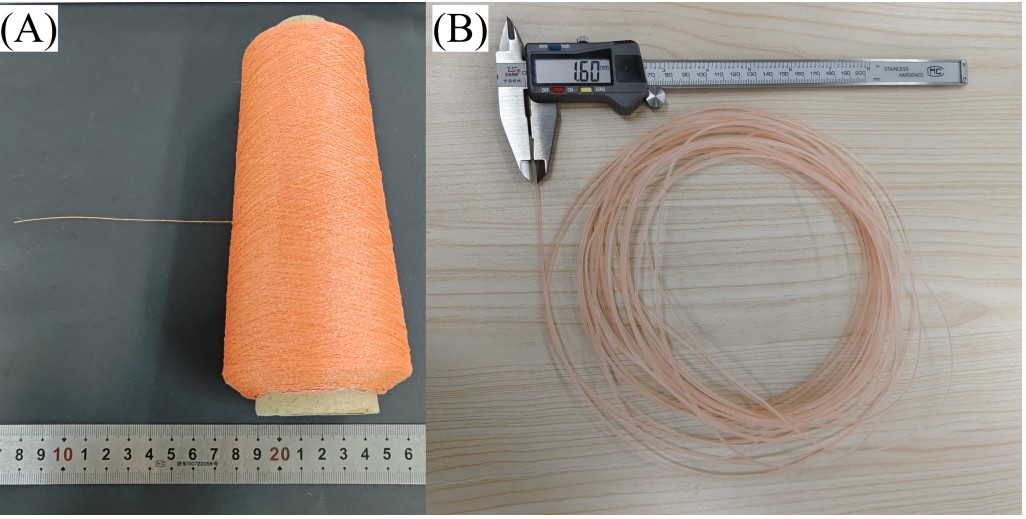

**Figure 2** Flax thread fiber and prepreg filaments.

Table 1 Fiber prepreg process parameters.

| Extrusion speed (r/min) | Traction speed (mm/min) | Diameter of die (mm) | Extrusion temperature (° C) | Impregnation temperature (° C) |
| --- | --- | --- | --- | --- |
| 8 | 300 | 1.5 | 145, 150 155, 160 | 140, 145 150, 155 |

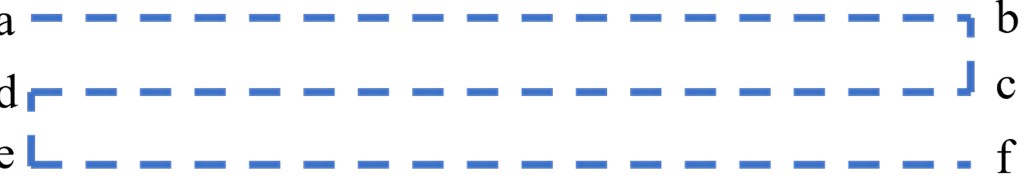

**Figure 3** CFF/PE composites molding path.

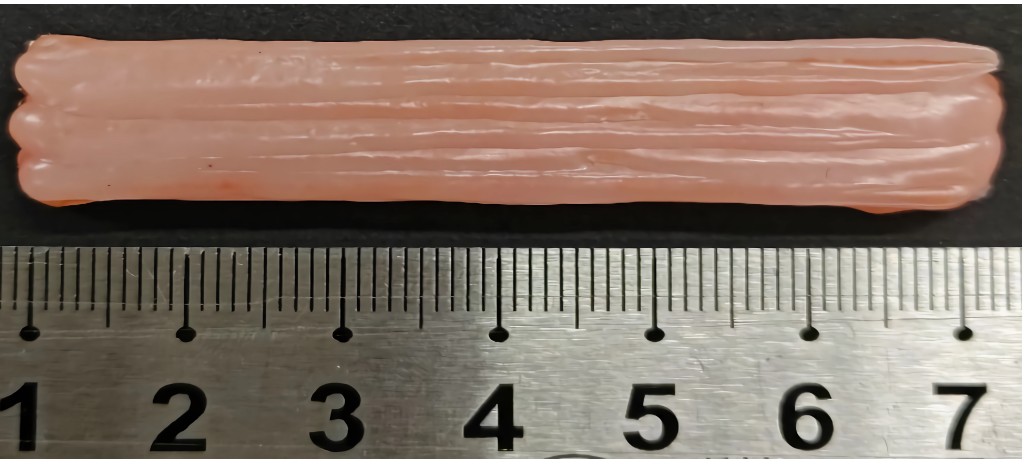

**Figure 4** Flexural specimen.

### 3D printing of composites

Due to the impossibility to achieve the continuity of continuous fiber without cutting by using the slicing software of 3D printers, the G-code was reedited to meet the experimental requirements. The printing path is illustrated in Fig. 3, with the printing order a→b→c→d→e→f. Then the CFF/PE prepreg filament was applied to a 3D printer (model 200, Allcct) to prepare flexural samples. The flexural specimen was shown in Fig. 4.

To enhance the print quality and accuracy of printed samples, preliminary printing tests were conducted to systematically evaluate key parameters, including printing temperature, printing speed, layer width, and layer height, aiming to determine the appropriate printing parameters for CFF/PE composites.

(1) Due to the poor adhesion of polyethylene (PE) to the print platform, CFF/PE composites are prone to being dragged by the print nozzle because of insufficient adhesion
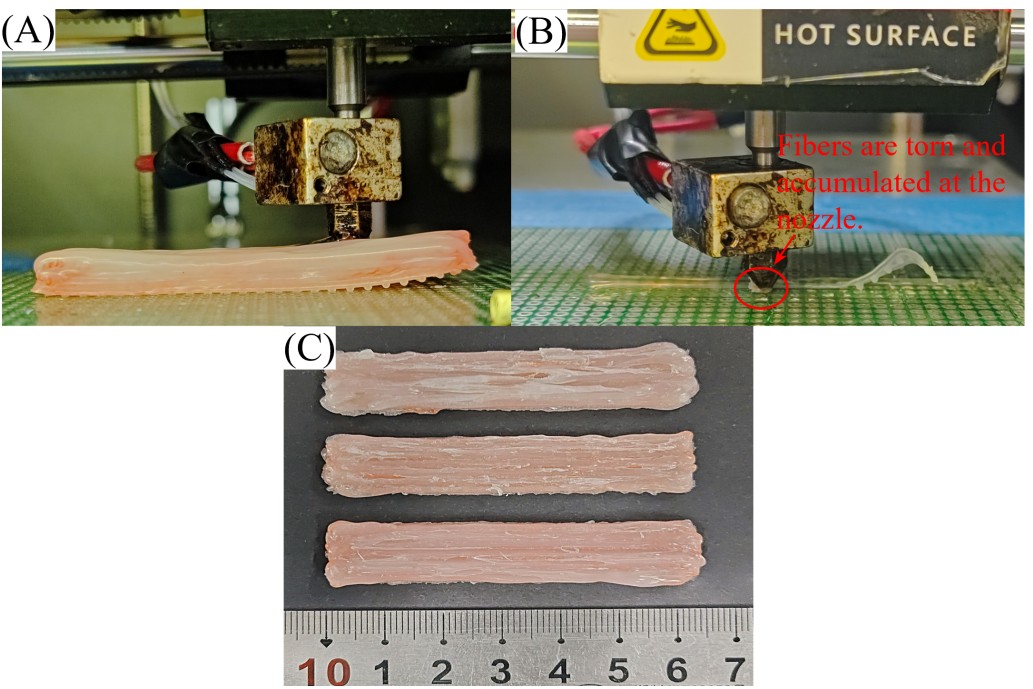

**Figure 5** CFF/PE prepreg filament printing: (A) warping phenomenon; (B) fiber was torn off; (C) surface quality defects.

to the print bed. Therefore, a porous plate was adopted to replace the platform of the 3D printer, CFF/PE composites exhibit better adhesion to the porous plate.

(2) To determine the minimum printing temperature, various temperatures were tested. When the printing temperature was below 180 °C, the printed parts tended to warp and curl (Fig. 5A). Therefore, three distinct printing temperatures were designed above 180 °C.

(3) To determine the appropriate printing speed, speed of 10, 15, 20 mm/min was explored. When the printing speed was 60 mm/min, issues such as fiber tearing, and accumulation leading to nozzle clogging were found (red circle in Fig. 5B). Figure 5C shows the samples printed at the same speed. It can be seen that the surfaces of the samples exhibited significant indentation and damage. The continuous flax fibers had been torn and were no longer visible. When the printing speed was below 10 mm/min, the longer staying of composites in nozzle, and further resulted in the degradation of PE and negatively impacted printing quality.

(4) The parameter design of layer width and thickness exerted significant impacts on the mechanical properties and surface quality of specimens, as comprehensively discussed in the subsequent sections titled 'Porosity of the composites' and 'Dimensional accuracy'.

Through trial and error, the final processing parameters for forming the CFF/PE composites were determined in Table 2.

**Table 2  CFF/PE composite FDM molding process parameters.**

| Main process parameters | Range of values | Other process parameters |
|---|---|---|
| Print temperature ($T_p$)/°C | 190, 200, 210 | $V = 10, H = 0.8, W = 2.0$ |
| Print velocity (V) mm/min | 10, 15, 20 | $T_p$=200, $H = 0.8, W = 2.0$ |
| Layer thickness (H) /mm | 0.6, 0.8, 1.0 | $T_p$=200, $V = 10, W = 2.0$ |
| Layering width (W) /mm | 1.5, 2.0, 2.5 | $T_p$=200, $V = 10, H = 0.8$ |

## Testing and characterization
### Tensile tests of prepreg filaments
According to the GB/T 228.1-2021 standard, the tensile properties of CFF/PE prepreg filaments affected by extrusion temperature and impregnation temperature were investigated by using a universal testing machine (UTM5105, Sansi). The process of the tensile test and the jigs were shown in Fig. 6. The test speed was set as five mm/min. Each group consisted of three test specimens, and the results were averaged.

### Mechanical properties of composites
To investigate the influence of FDM process parameters on the bending properties of CFF/PE composites, bending tests were conducted using an electronic universal testing machine (UTM5105, Sansi). The specimen dimensions were 65 mm ×12 mm ×6 mm, with a span of 40 mm and a testing speed of 10 mm/min. After the bending angle of the being tested specimens remains constant, repeated experiments are conducted five times under the same conditions to explore the influence of experimental repetitions on flexural strength and rebound angles.

### Porosity
The calculation of the porosity ($V_v$) of the composites was performed using the density measurement method (*Li, 2012*) with the following formula:

$$V_v = 100 - \rho_p \left( W_r / \rho_r + W_f / \rho_f \right) \tag{1}$$

$$\rho_p = m_p / V_p. \tag{2}$$

Where, $\rho_p$ is the density of the composites, $W_r$ is the resin mass fraction, $\rho_r$ is the resin density, which is 0.93 g/cm³, $W_f$ is the fiber mass fraction, $\rho_f$ is the fiber density, which is 1.5 g/cm³, $m_p$ is the mass of the composites, and $V_p$ is the volume of the composites.

### Morphology characterization and melt flow index test
The tensile fracture surfaces of the prepreg filaments were coated with gold, and then were observed using a scanning electron microscope (VEGA 3 SBH, Czech) to examine the microstructure and the interfacial bonding between the continuous fibers and the resin.

According to the experimental standard of GB3682-2000, measure the melt index of PE using a melt index tester (ST-400B, EAST) with a standard load of 2.16 kg.

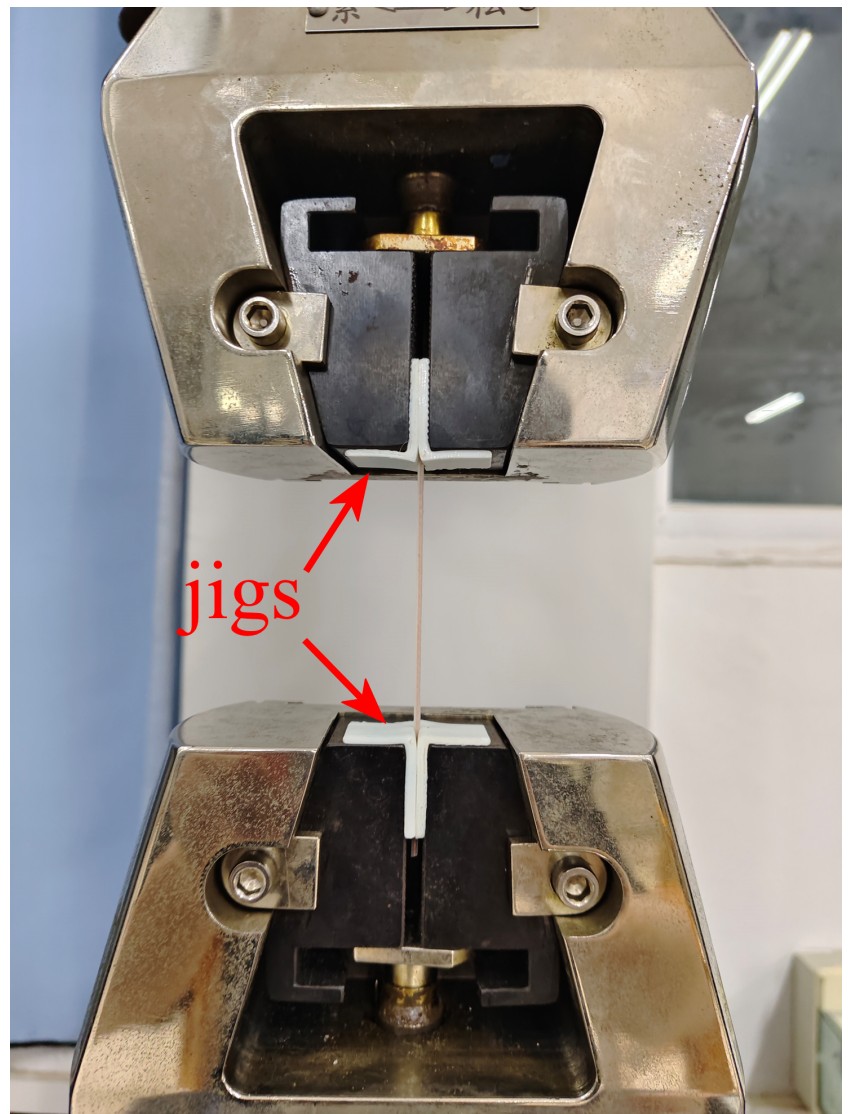

**Figure 6  Process of tensile test for prepreg filaments and the jigs.**

## RESULTS AND DISCUSSION

### Melt flow index of PE

The melt flow index (MFI) reflects the fluidity of PE. As shown in Fig. 7, during the process of temperature rising from 140 °C to 160 °C, the MFI gradually increased from 16.26 g/10 min to 26.94 g/10 min. The increase in temperature led to enhanced mobility of PE molecular chains, weakened intermolecular interactions, reduced viscosity of PE, and increased fluidity (*Güldaş et al., 2018*).

### *Effect of temperature on the tensile properties of the prepreg filaments*

Figure 8 illustrates the effect of different temperatures on the tensile properties of CFF/PE prepreg filaments. As the temperature increased from 140 °C–145 °C to 150 °C–155 °C,

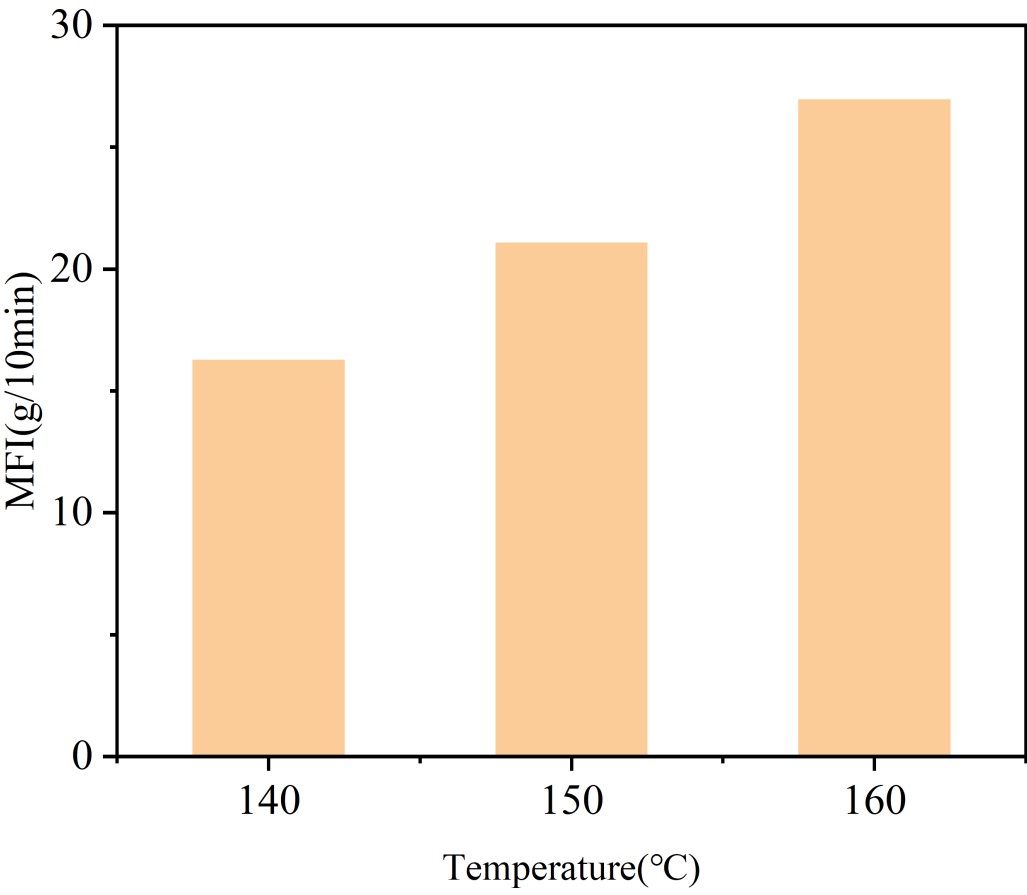

**Figure 7  Melt flow index of PE at different temperatures.**

the tensile strength rose from 10.31 MPa to 12.41 MPa. Further increases in temperature resulted in a decrease in tensile strength to 11.03 MPa. Compared to pure PE filament at 155 °C (7.27 MPa), the maximum tensile strength of CFF/PE composites has increased by approximately 70.7%.

At temperature of 140 °C–145 °C, the flowability of PE was common (Fig. 7), resulting in only a small portion of the PE coating the flax fibers (red arrow in Fig. 9A), and the interior of the flax fibers remained uncoated by PE (yellow arrow in Fig. 9A). During the tensile, these un-impregnated fibers were stretched after the coated PE has fractured. This phenomenon indicated poor interfacial compatibility between PE and CFF. With increasing temperature, the impregnate degree gradually enhanced, the impregnated fibers needed to consume more energy to be stretched from the PE matrix, resulting in higher strength.

At temperatures of 150 °C–155 °C, most of the PE impregnated into the flax fiber interiors (red arrow in Fig. 9C), with only a flax few fibers being stretched (yellow arrow in Fig. 9C), resulting in the best tensile properties for the prepreg filaments at this temperature. However, continuously increased temperature to 155 °C–160 °C, the

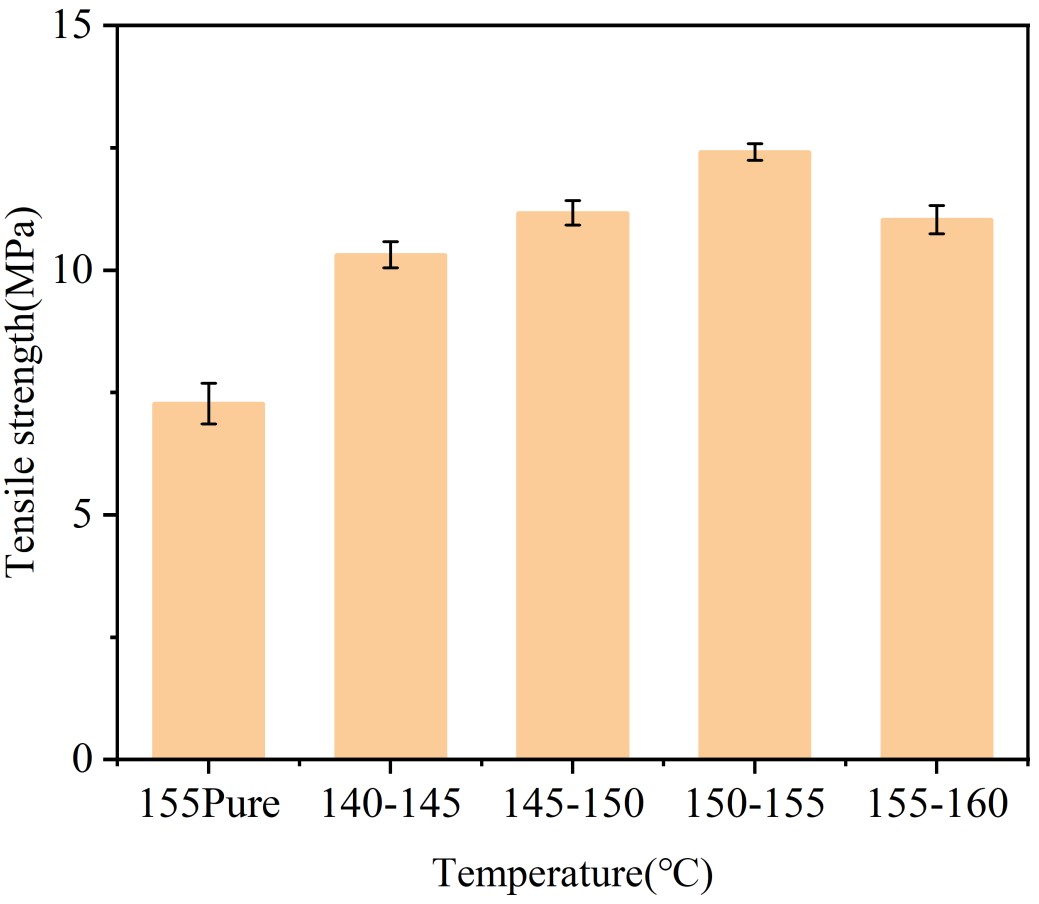

**Figure 8** **Effect of different temperatures on the tensile strength of CFF/PE prepreg filaments.**

tensile fracture surfaces showed uniformity (Fig. 9D), the interfacial bonding in CFF/PE did not incur additional energy expenditure during the failure process, consequently leading to a reduction in the tensile properties of the prepreg filaments.

## Flexural properties of composites
### Effect of layer width

Modifying the layer width equivalents to adjusting the distance between adjacent deposition lines. Figure 10 shows the stacking theory model, where lines overlapped (red arrow) when the layer width was smaller than line width. Proper overlapping meant higher dimensional accuracy and improved mechanical properties of the printed parts. However, excessive overlapping may negatively impact print quality and strength of the parts (*Dönitz et al.,* *2023*). Figure 11 shows the stress–strain curves of CFF/PE composites in various layer width and specimens post-bending test. (A)–(D) represent the maximum values under different parameter. As the layer width decreased from 2.5 mm to 1.5 mm, the flexural strength increased from 14.49 MPa to 24.74 MPa, with an enhancement of 41.43%. Compared to pure PE samples at 1.5 mm, there was an increase of about 16.33%. The rebound angle of the bend specimen also decreased from 160° to 149°. The increase in layer width resulted in

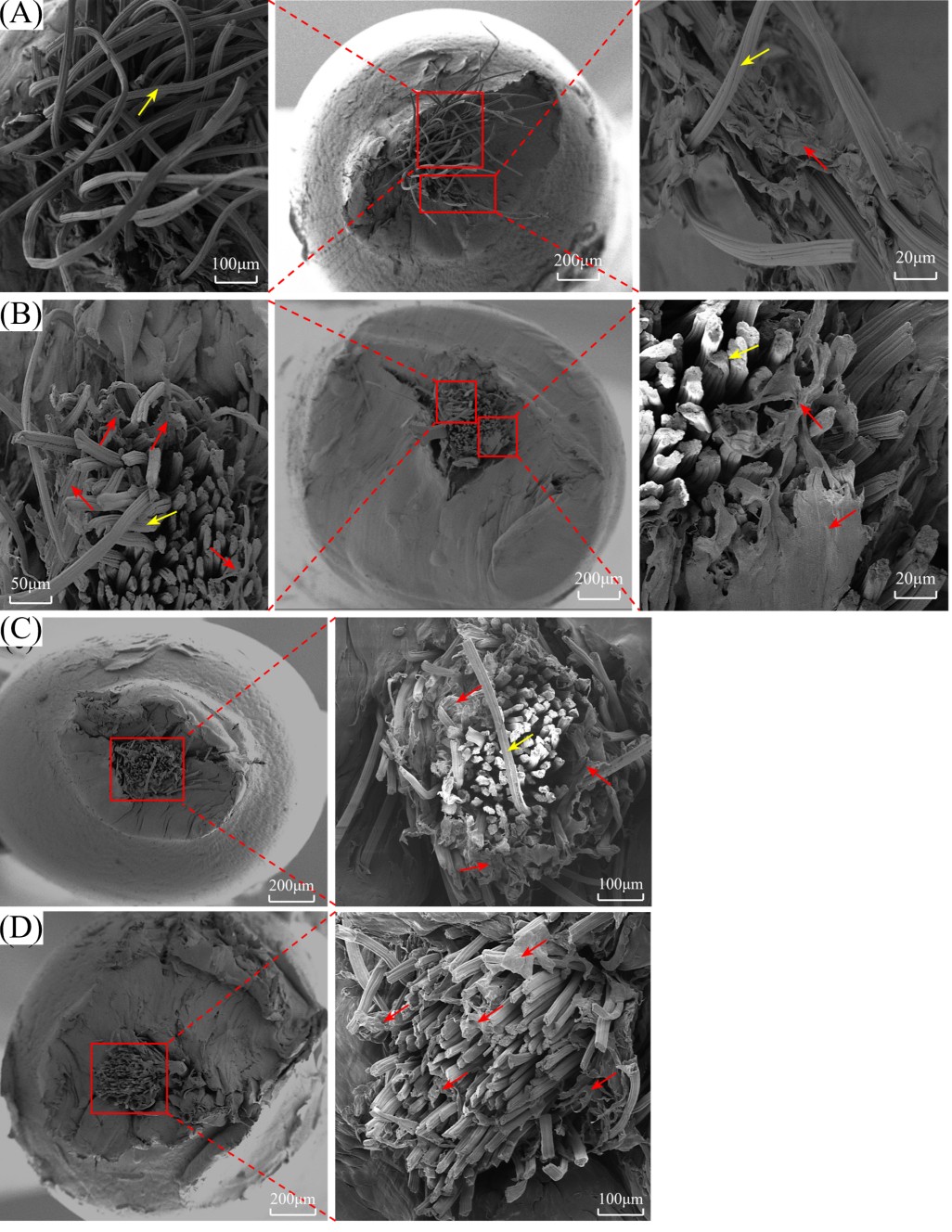

**Figure 9** Microstructures of the tensile fracture surface of CFF/PE prepreg filaments: (A) 140 °C–145 °C; (B) 145 °C–150 °C; (C) 150 °C–155 °C; (D) 155 °C–160 °C; (The red arrow indicates the PE and the yellow arrow indicates the flax thread fiber).

reduced overlapping between fibers, leading to diminished bonding effects. Under external bending forces, stress could not be effectively transmitted, thereby weakening the bending performance of the composites, and also reducing the rebound angle of the specimen.

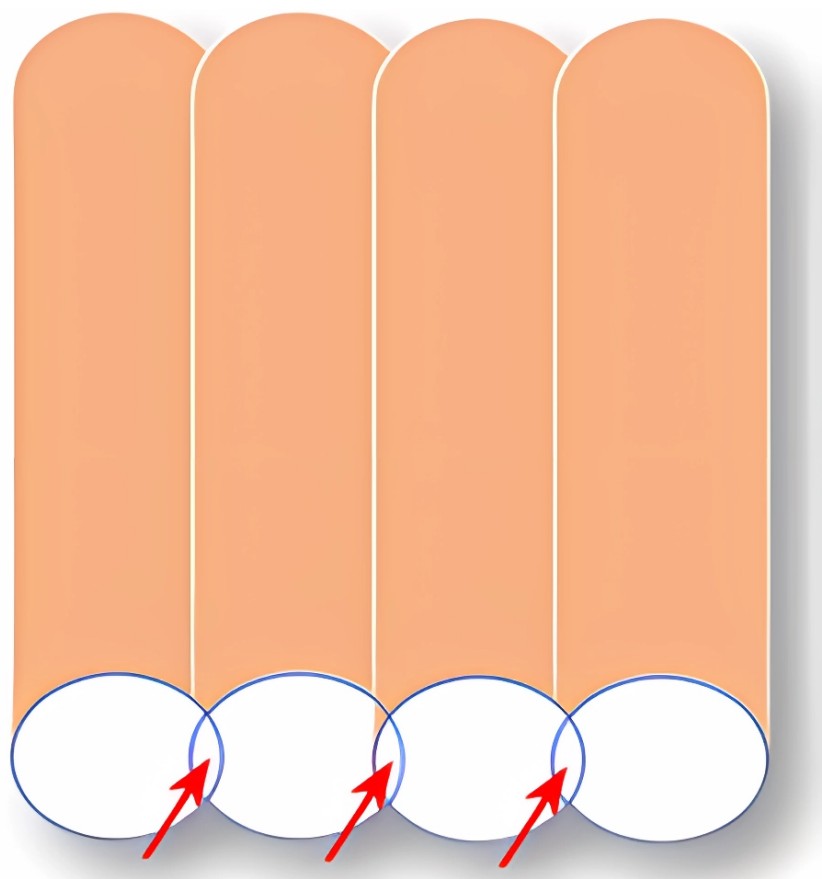

**Figure 10** CFF/PE composite stacking model diagram (arrows indicate stacking regions).

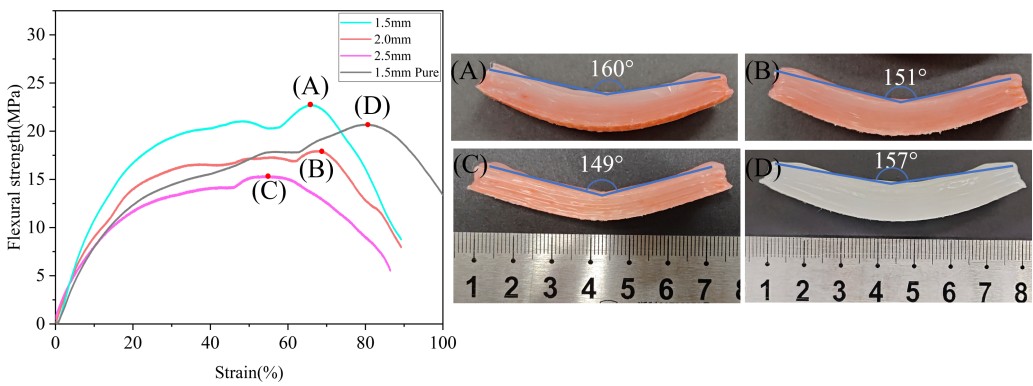

**Figure 11** Stress–strain curves of CFF/PE composites with different layer width and specimens post-bending.

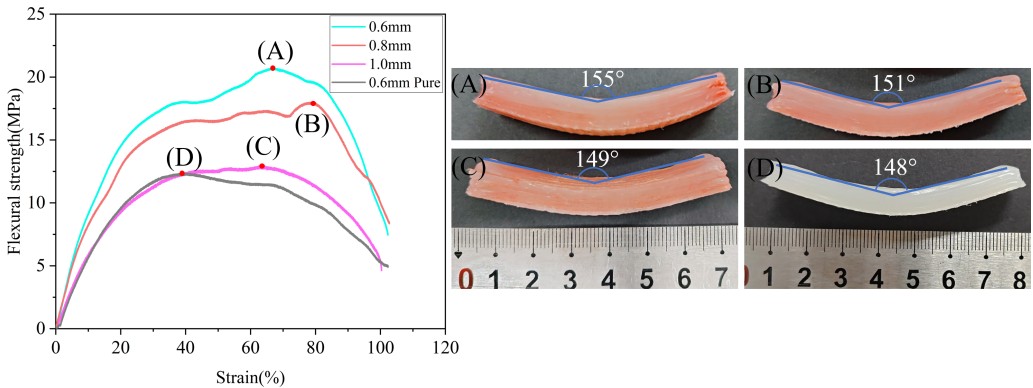

**Figure 12 Stress–strain curves of CFF/PE composites with different layer thicknesses and specimens post-bending.**

### Effect of layer thickness

Figure 12 shows the stress–strain curves of CFF/PE composites in various layer thicknesses and specimens post-bending test. (A)–(D) represent the maximum values under different parameter. The increase in layering thickness from 0.6 to 1.0 mm resulted in a reduction of flexural strength from 20.94 MPa to 13.14 MPa, and the rebound angle after bending of the specimen also decreased from 155° to 149°. This phenomenon can be contributed to that the increasing layer thickness reduced the interlayer pressure, further weakened the interfacial bonding strength between layers and decreased the flexural performance. At the same time, it also caused the rebound angle after bending to decrease. However, compared to pure PE printed at a layer thickness of 0.6 mm, the maximum flexural strength increased by 69.55%.

### Effect of print temperature

Figure 13 shows the stress–strain curves of CFF/PE composites in various printing temperatures and specimens post-bending test. (A)–(D) represent the maximum values under different parameter. When the printing temperature increased from 190 °C to 200 °C, the flexural strength rose from 18.40 MPa to 19.61 MPa. Further increasing the printing temperature to 210 °C resulted in a decrease in flexural strength to 18.54 MPa. Within the investigated temperature range, an increase in temperature improved the interfacial bonding between PE phases, enhanced the mechanical properties of the composites. However, when the temperature was raised further, the reduced viscosity of the PE matrix adversely affected printing quality, leading to a reduction in mechanical performance. Additionally, the rebound angle of the specimens after bending varied only within a very small range.

### Effect of 3D printing speed

Figure 14 shows the stress–strain curves of CFF/PE composites in various printing speeds and specimens post-bending test. (A)–(D) represent the maximum values under different parameter. The variations in printing speed for the flexural properties were not obvious for

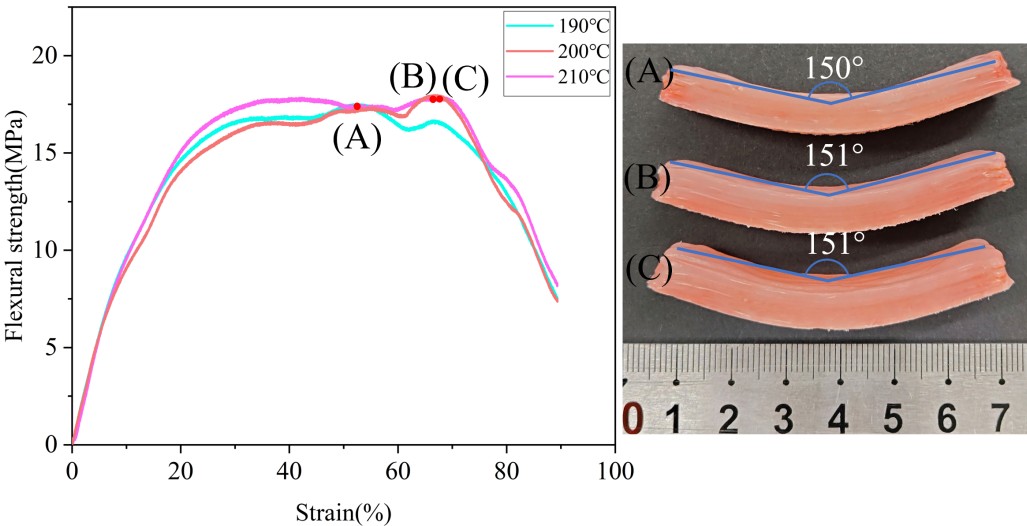

**Figure 13** Stress–strain curves of CFF/PE composites at different print temperatures and specimens post-bending.

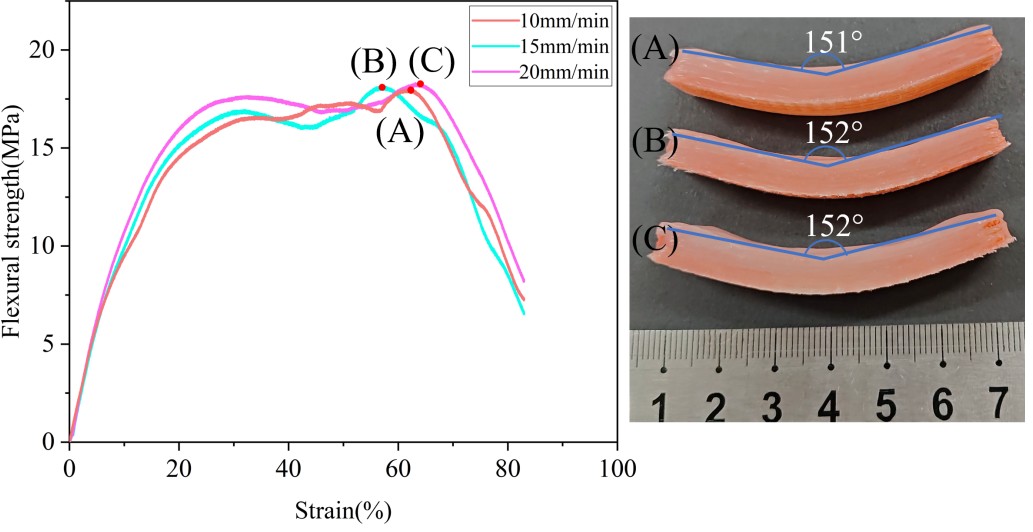

**Figure 14** Stress–strain curves of CFF/PE composites at different print speeds and specimens post-bending.

the CFF/PE composites. The rebound angle of the specimens after bending was also not noticeable. This can be contributed to the small range of printing speed, In section "3D printing of composites", printing speed was 60 mm/min resulted in the fiber breakage and poor adhesion (Figs. 5B, 5C), and below 10 mm/min significantly reduced the printing efficiency. Therefore, the printing speed should be within the range of 10–20 mm/min.

**Table 3  Effects of different process parameters on the porosity of CFF/PE composites.**

| Process parameters | | Porosity/% | Flexural strength/MPa |
|---|---|---|---|
| Constants/mm | Variables/mm | | |
| W=2.0 | $H = 0.6$ | 3.88 | 20.94 |
| | $H = 0.8$ | 5.74 | 19.61 |
| | $H = 1.0$ | 11.66 | 13.14 |
| H=0.8 | $W = 1.5$ | 4.75 | 24.74 |
| | $W = 2.0$ | 5.74 | 19.61 |
| | $W = 2.5$ | 13.27 | 14.49 |

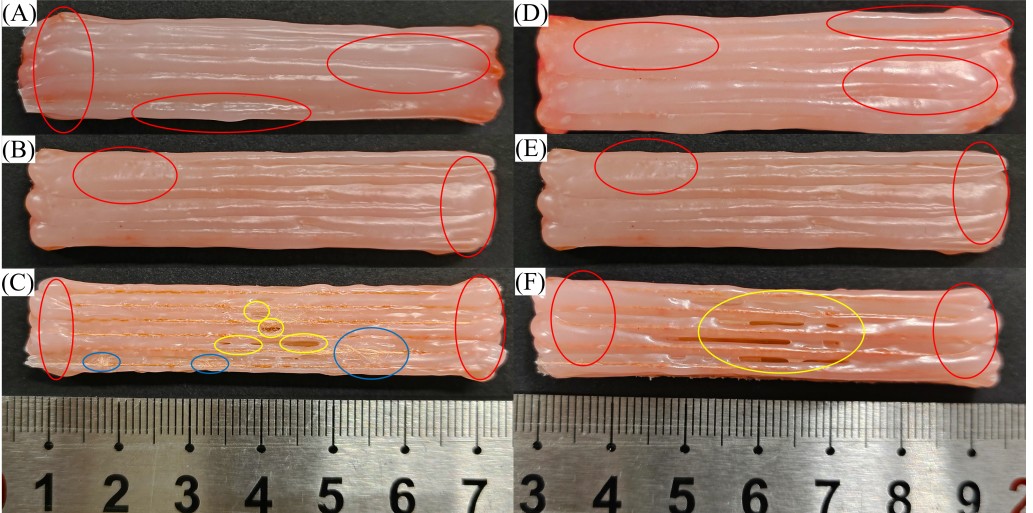

**Figure 15  Surface conditions of CFF/PE composites:** (A, B, C) shows the layer thicknesses of 0.6 mm, 0.8 mm, and 1.0 mm, respectively; (D, E, F) shows the layer width of 1.5 mm, 2.0 mm, and 2.5 mm, respectively. (Red circles indicate surface resin accumulation, blue circles indicate fiber detachment, and yellow circles indicate voids).

## Porosity of the composites

The layer width and layering thickness are critical factors affecting the porosity of the composites. Changes in these parameters directly altered the voids between deposition lines and the bonding strength between layers, thereby influencing the mechanical properties of the composites (*Cheng et al., 2021*). Table 3 shows a positive correlation between porosity and processing parameters, and a negative correlation with flexural strength. Figure 15 illustrates the surface conditions of CFF/PE composites under various processing parameters.

When the layer thickness increased from 0.6 mm to 1.0 mm, the surface of the CFF/PE composites exhibited a few voids (yellow circles in Fig. 15C), and fiber broke (blue circles in Fig. 15C). The increase in layer thickness led to a weakening of the adhesion between layers, which hindered the effective bonding between them (*Tian et al., 2021*). Friction

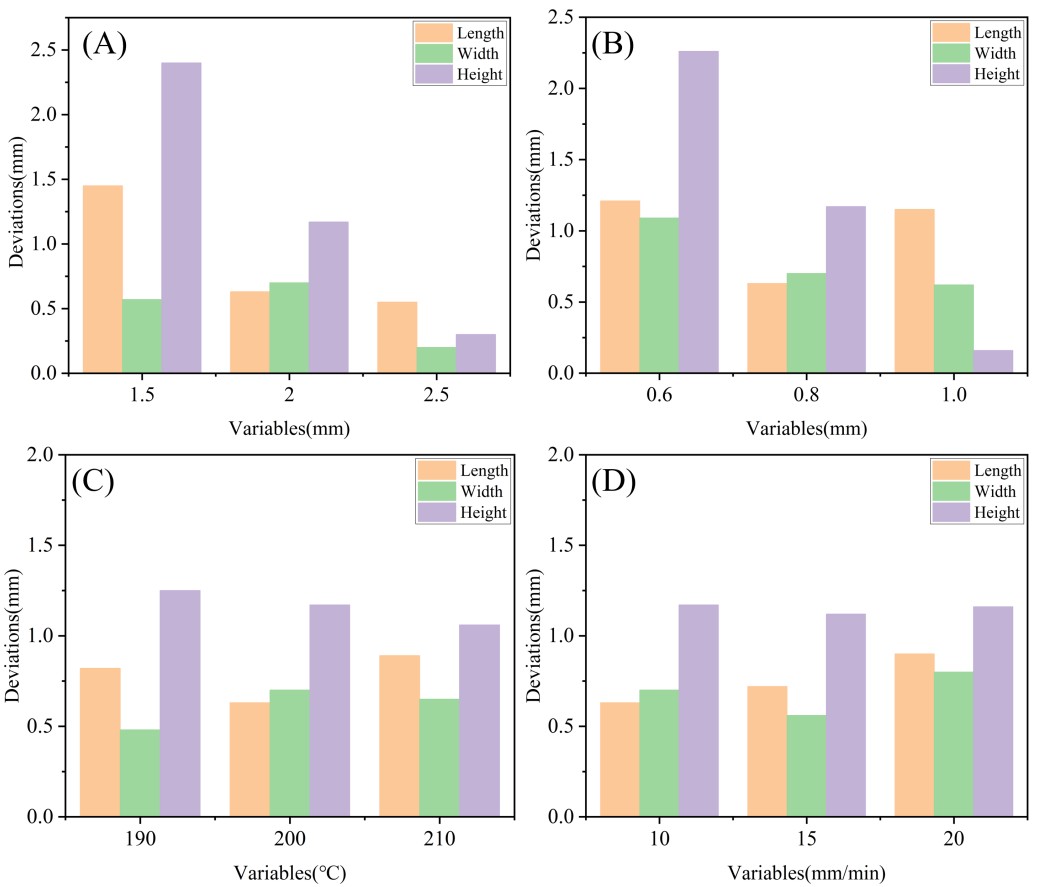

**Figure 16** **Dimensional deviation diagram of CFF/PE composites under different process parameters: (A), (B), (C), and (D) represent layer width, layer thickness, print temperature, and print speed, respectively.**

between fibers and the nozzle during the printing process ultimately led to the creation of voids and the fracture of fibers.

When the layer width was increased from 1.5 mm to 2.5 mm, a large number of voids appeared on the surface of the CFF/PE composites (yellow circles in Fig. 15F). The increase in layer width led to an enlargement of the gaps between adjacent deposited lines, reduced the bonding effects, and further resulted in an increase in porosity.

For CFF/PE composites, the larger the porosity, the smaller the bending strength. When the layering thickness increased from 0.6 mm to 1.0 mm, porosity rose from 3.88% to 11.66%, and flexural strength decreased from 20.94 MPa to 13.14 MPa. Similarly, when the layer width increased from 1.5 mm to 2.5 mm, porosity increased from 4.75% to 13.27%, and flexural strength decreased from 24.74 MPa to 14.49 MPa.

## Dimensional accuracy

Figures 16 and 17 illustrate the dimensional deviation and surface conditions of CFF/PE composites under different process parameters, respectively. When the layer width was

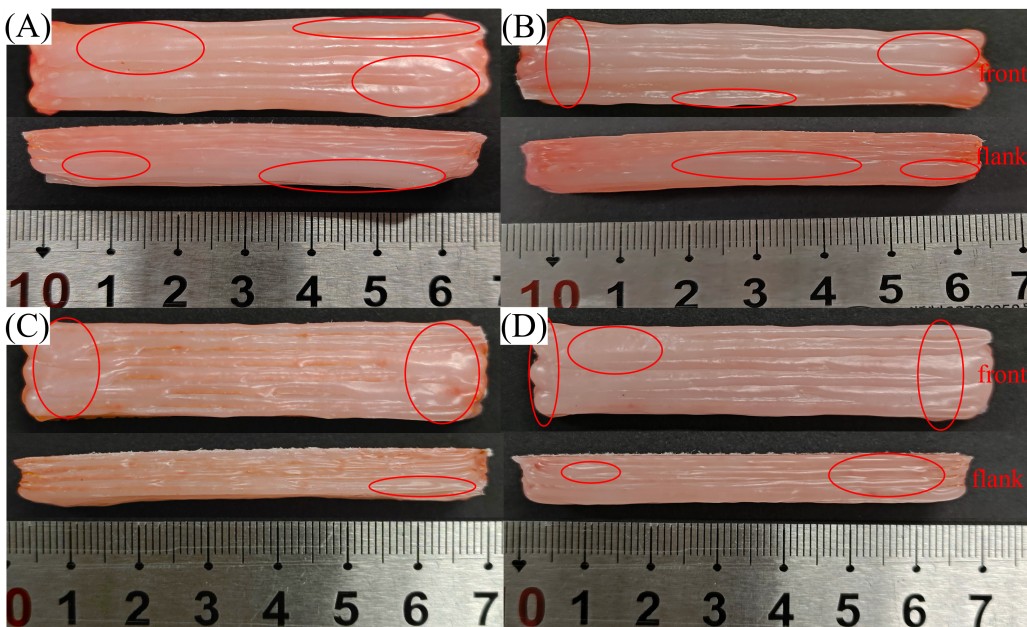

**Figure 17  PE accumulation on the surface of CFF/PE composites: (A), (B), (C), and (D) represent layer width, layer thickness, print temperature, and print speed, respectively; severe PE accumulation within red circles.**

2.5 mm, the front and side surfaces of the composites exhibited a lot of PE accumulation (red circles in Fig. 17A), especially at the edges, where the composites exhibited the maximum dimensional error group of $1.45 \times 0.57 \times 2.32$ mm (as shown in Fig. 16A). When the layer thickness was 0.6 mm, the thinner layer caused excess PE melts to be squeezed to the edges of the specimen during printing (red circles in Fig. 17B), leading to an increase in dimensional error of $1.21 \times 1.09 \times 2.26$ mm (as shown in Fig. 16B).

The impact of printing temperature and speed on the dimensional accuracy of the composites was relatively small, with printing errors around one mm. As the printing temperature increased from 190 °C to 210 °C, the dimensional error of the printed CFF/PE parts increased to the maximum deviation reaching of $0.89 \times 0.65 \times 1.25$ mm (as shown in Fig. 16C). This could be attributed to the reduction in resin viscosity at higher temperatures. The change in printing speed has a similar effect on the length of the composites. A printing speed of 20 mm/min took a significant impact on the width of the specimen, which was due to the instability of the extrusion amount as the printing speed increasing (as shown in Fig. 16D).

### Effect of the number of bending on the flexural properties of composites

Figures 18 shows the stress–strain curves of CFF/PE composites after five flexural tests in various processing parameters. It can be observed that (A), (B), (C), and (D) all exhibited a similar pattern. With an increase in the number of experiments, the flexural strength of CFF/PE composites gradually decreased firstly and kept stability after the third experiment.

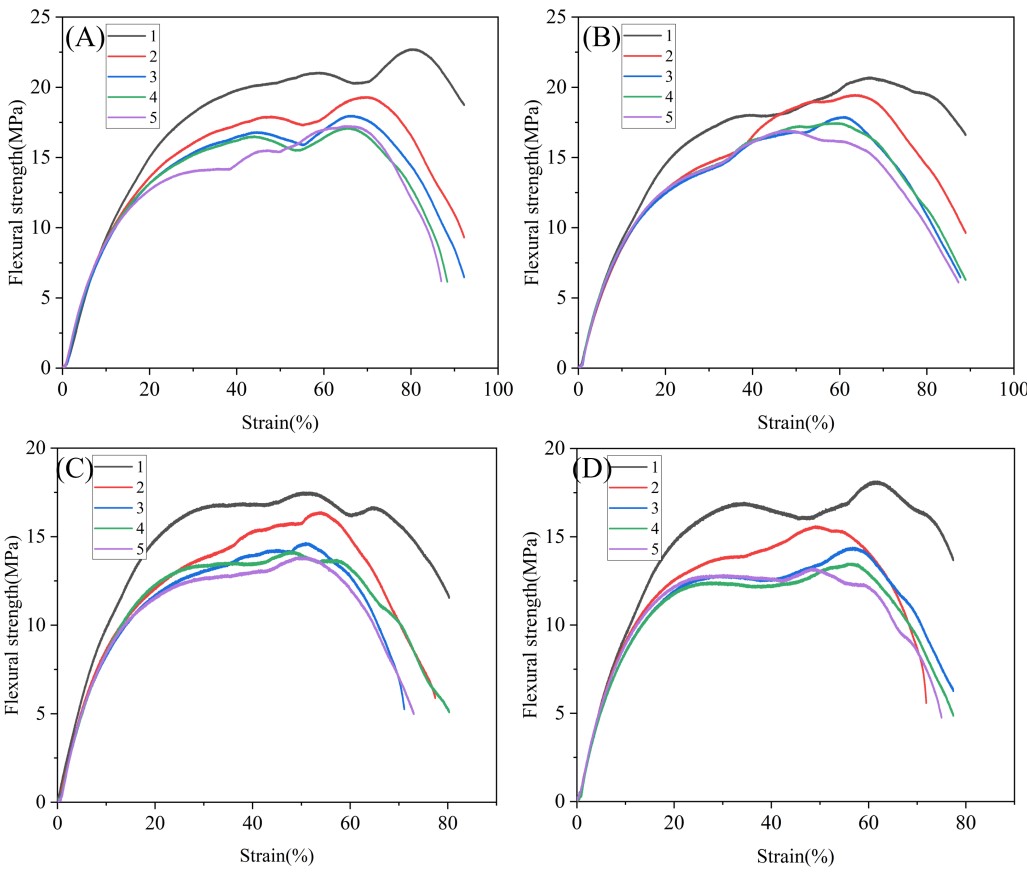

**Figure 18** Stress–strain curves of CFF/PE composites after five flexural tests under different parameters: (A), (B), (C), and (D) represent layer width , layer thickness, print temperature, and print speed, respectively.

As shown in Fig. 18A, with a layer width of 1.5 mm, the flexural strength decreased from 24.74 MPa in the 1st test to 20.75 MPa in the 2nd test, and further to 19.68 MPa in the 3rd test. The extent of decrease in flexural strength diminished with increasing test numbers, reaching 19.06 MPa in the 4th test and showing no change in the 5th test. Samples under different layering thicknesses, printing temperatures, and printing speeds generally exhibited the same trend, with flexural strength to keep stability by the 4th test (Figs. 18B, 18C, 18D).

Figure 19 shows the bending behavior of CFF/PE composites. During the bending processing, the composites undergone elastic deformation and plastic deformation. When the bending force was removed, the elastic deformation recovered and the composite samples rebounded. However, with the increasing the bending number, the degree of elastic deformation decreased, resulting in the decreasing rebound angle. Shown in Fig. 19, during the first three bending experiments, the rebound angle reduced from 162° to 151° and kept level to 150° in the following bending.

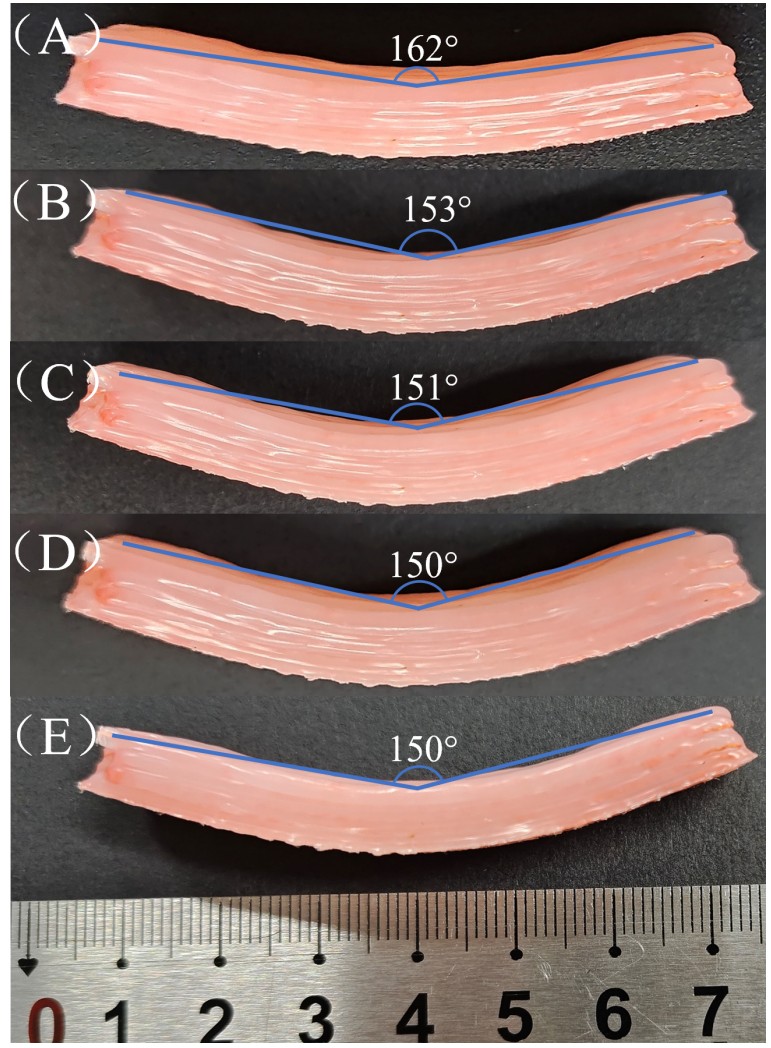

**Figure 19** Flexural rebound behavior of CFF/PE Composite: (A–E) represent rebound angles for 1st to 5th bending tests, respectively.

## CONCLUSIONS

(1) This paper used a home-made single-screw extrusion device to produce continuous flax fiber (CFF) reinforced polyethylene (PE) prepreg filaments. The effects of different extrusion and impregnation temperatures on the tensile properties of CFF/PE composite filaments were investigated. The best impregnation effect and the highest tensile strength of 12.41 MPa were achieved at temperature ranging from 150 °C to 155 °C. Compared to pure PE filament at 155 °C (7.27 MPa), the maximum tensile strength of composite filament has increased by approximately 70.7%, indicating that CFF enhanced the tensile strength dramatically.

(2) Using CFF/PE prepreg filaments as raw material, the effects of FDM process parameters on the mechanical properties of printed samples were investigated. When the

layer width was 1.5 mm, the layer thickness was 0.8 mm, the print temperature was 200 °C and the print speed was 10 mm/min, the flexural strength of FDM CFF/PE composites was maximized at 24.74 MPa, which was higher about 16.33% than that of pure PE.

(3) The printing parameters: layer width (W) and layer thickness (H) exerted significant influence on the porosity of FDM CFF/PE composites. When W and H were 2.0 mm and 0.6 mm respectively, the porosity of the FDM CFF/PE composite reduced to the valley with value of 3.88%.

(4) The layer width (W) and layer thickness (H) had a significant impact on the dimensional accuracy of FDM CFF/PE composites. When W and H were 1.5 mm and 0.8 mm respectively, the dimensional error of the FDM CFF/PE composite reached to the maximum with error of $1.45 \times 0.57 \times 2.32$ mm.

(5) In order to investigate the effects of bending cycles on the rebound angle and bending performance of CFF/PE composites, five bending experiments were conducted. As the number of bending increased, both the bending rebound angle and flexural strength decreased, with the extent of decrease negatively correlated to the number of experiments. Both parameters stabilized during the 4th bending experiment.

The CFF/PE composites produced in this study exhibited superior mechanical properties compared to the pure PE, and the use of natural fibers embodied the principles of sustainable development. The natural fiber-reinforced PE composites manufactured using 3D printing technology possess broad application prospects in the future manufacturing industry.

### Funding

This work was supported by the institution of National Natural Science Foundations of China (52365043), Natural Science Foundations of Guangxi province (2023JJA160223, 2023JJA160056), Middle-aged and Young Teachers' Basic Ability Promotion Project of Guangxi (2020KY21012) and Guangxi Science and technology planning project (2023AB01248). The funders had no role in study design, data collection and analysis, decision to publish, or preparation of the manuscript.

### Grant Disclosures

The following grant information was disclosed by the authors:
National Natural Science Foundations of China: 52365043.
Natural Science Foundations of Guangxi province: 2023JJA160223, 2023JJA160056.
Middle-aged and Young Teachers' Basic Ability Promotion Project of Guangxi: 2020KY21012.
Guangxi Science and technology planning project: 2023AB01248.

### Competing Interests

The authors declare there are no competing interests.
## Author Contributions

- Minggan Wang conceived and designed the experiments, performed the experiments, analyzed the data, performed the computation work, prepared figures and/or tables, authored or reviewed drafts of the article, and approved the final draft.
- Xiaohui Song conceived and designed the experiments, analyzed the data, performed the computation work, prepared figures and/or tables, authored or reviewed drafts of the article, and approved the final draft.
- Zhengwei Yang performed the experiments, authored or reviewed drafts of the article, and approved the final draft.
- Chunlei Luo analyzed the data, authored or reviewed drafts of the article, and approved the final draft.
- Songda Chi performed the computation work, authored or reviewed drafts of the article, and approved the final draft.
- Hao Jiang analyzed the data, authored or reviewed drafts of the article, and approved the final draft.

## Data Availability

The raw data are available in the Supplemental File.

## Supplemental Information

Supplemental information for this article can be found online at http://dx.doi.org/10.7717/peerj-matsci.34#supplemental-information.

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
