# Peer review of "Flexural properties of 3D printed continuous flax fiber reinforced polyethylene composites"

_PeerJ Materials Science, doi:10.7717/peerj-matsci.34_

## Round 0.1 · original submission · Major Revisions

Dear authors, based on the reviewers' comments, your manuscript needs a major revision. Please carefully check each comment and revise your manuscript accordingly.

Reviewer 1 ·

Basic reporting

No Comment

Experimental design

The authors have designed an instrument for the preparation of continuous flax fiber-reinforced polyethylene filaments. There is some description of this device in the materials and methods section along with a diagram in Figure 1. However, this information is insufficient to guide others in the preparation of similar devices. The authors need to add more detail on this experimental setup.

Validity of the findings

The authors are very thorough in their experiments and analysis. There is one figure that I think would help to support their findings. I would like to see an SEM of the cross-section of a fiber so one could see the full 1.6 mm fiber.
This might be appropriate as an extra set of panels in Figure 9. As figure 9 stands, it is very difficult for me to tell an unimpregnated fiber from an impregnated fiber (outside of the arrows that the authors use). However, the reader should be able to ascertain this information with a better composed SEM image.

Reviewer 2 ·

Basic reporting

The article is interesting, and the reported results are coherent.

Experimental design

The results are based on the fabrication of the composite using a "self developed single screw extruder".
This machine is not described anywhere in the paper and it is important for the repeatability / reproducibility of the research. I would suggest to add a section with the description and the information for building this single screw extruder.

Validity of the findings

Please see my previous point on the single-screw extruder.
It's not clear how the authors detect if the fibers have been impregnated or not (figure 9 and in the main paper). Can you please describe how you determine the impregnation of the fibres?

Reviewer 3 ·

Basic reporting

L103: Is the flax density provided by the supplier or determined experimentally by the authors? If the latter, the method should be described since flax density in most literature ranges between 1.4 and 1.5 g/cm³.
L135: The term "nozzle dragging" requires further explanation.
L139: The authors describe using a trial-and-error method to determine processing parameters. They should:
Clearly list and define all parameters upfront.
Detail how they optimized these parameters, particularly the "scan spacing," which is unconventional and should perhaps be referred to as "layer width."
L154: Additional information is needed about the cycle bending tests, specifically how and when the authors determine the test is complete.
L175, L204: Use consistent figure labeling throughout (e.g., "Figure 7" instead of "fig. 7").
The English language throughout the article needs significant improvement.

Experimental design

There are significant issues with the selection of parameters and the background research conducted:
Printing Speed:
The units for printing speed seem incorrect. Did the authors mean 10, 15, and 20 mm/s instead of mm/min? If the latter, the speeds are exceptionally slow, even for continuous flax fiber printing. For comparison, Yung et al. [2] explored speeds of 5, 10, and 15 mm/s and reported no fiber rupture issues at 15 mm/s. If the speeds are indeed in mm/s, the authors should still consider including a lower speed like 5 mm/s, as it has been shown to yield better mechanical properties. Efficiency concerns are not a valid justification for excluding lower speeds, particularly when mechanical performance is a priority.
Layer Height and Width:
The authors treat layer height and layer width as independent parameters, which is problematic when using a pre-impregnated filament. Since the filament's cross-sectional area (volume of PLA and fiber) is fixed, the programmed strip (layer height × layer width) must match the filament's cross-sectional area.
Increasing one parameter (e.g., layer height) without adjusting the other injects porosity into the printed sample, as there will not be enough material to fill the gap. This highlights a lack of understanding of the pre-impregnation technique and suggests the authors need to review relevant literature.

Validity of the findings

L235: The authors state there is no significant difference in flexural strength, despite reporting an increase from 18.4 MPa to 19.61 MPa (a 6.6% improvement resulting from a 5.3% increase in nozzle temperature). This increase is not negligible and warrants further discussion. Why was it considered insignificant?

Additional comments

The article lacks sufficient references related to 3D-printed continuous flax fiber-reinforced polymers. Notably, several key works relevant to the subject are missing:
Le Duigou et al. [1]: This study examines the influence of slicing parameters on the mechanical properties of continuous flax fiber (CFF)/PLA printed composites, which aligns closely with the current work.
Yung et al. [2]: This study introduces the "recovery effect," which affects dimensional consistency in printed samples. The authors also investigate the influence of printing speeds on recovery and mechanical properties, concluding that 5 mm/s yields better results than 10 or 15 mm/s.
While these works focus on a different matrix (PLA), they provide valuable insights that could guide the authors in refining their study.
1. Le Duigou, A., et al., Tailoring the mechanical properties of 3D-printed continuous flax/PLA biocomposites by controlling the slicing parameters. Composites Part B: Engineering, 2020. 203: p. 108474.
2. Investigation of recovery behavior on 3D-printed continuous plant fiber-reinforced composites | Signed in. Additive Manufacturing, 2024. 88.

---

## Round 0.2 · accepted · Accept

Based on the reviewer´s comments, the revised version of this manuscript can be accepted as it is.

Reviewer 1 ·

Basic reporting

No Comment

Experimental design

The authors have answered my questions regarding experimental setup.

Validity of the findings

The authors have made appropriate additions to their findings (as suggested by all reviewers) and modified their discussion and analysis as well.

Reviewer 2 ·

Basic reporting

The manuscript has improved after this round of review.

Experimental design

The experimental part has improved and now the machine has a reference.

Validity of the findings

The manuscript has improved after this round of review.